# Effectiveness and Safety of Erector Spinae Plane Block vs. Conventional Pain Treatment Strategies in Thoracic Surgery

**DOI:** 10.3390/jcm14092870

**Published:** 2025-04-22

**Authors:** Bernhard Zapletal, Paul Bsuchner, Merjem Begic, Alexis Slama, Alexander Vierthaler, Marcus J. Schultz, Edda M. Tschernko, Peter Wohlrab

**Affiliations:** 1Department of Anesthesiology, General Intensive Care and Pain Medicine, Division of Cardiac Thoracic Vascular Anesthesia and Intensive Care Medicine, Medical University Vienna, 1090 Vienna, Austria; bernhard.zapletal@muv.ac.at (B.Z.); paul.bsuchner@gmail.com (P.B.); alexander.vierthaler@icloud.com (A.V.); marcus.j.schultz@gmail.com (M.J.S.); edda.tschernko@muv.ac.at (E.M.T.); 2Department of Anesthesiology and Intensive Care, Orthopaedic Hospital Vienna-Speising, 1130 Vienna, Austria; 3Department of Thoracic Surgery, Medical University Vienna, 1090 Vienna, Austria; merjem.begic@muv.ac.at (M.B.); alexis.slama@muv.ac.at (A.S.); 4Department of Intensive Care, Amsterdam University Medical Centers, 1105 AZ Amsterdam, The Netherlands

**Keywords:** video-assisted thoracic surgery, VATS, thoracotomy, pain, erector spinae plane block, ESPB, thoracic epidural analgesia, TEA, intravenous combination analgesia

## Abstract

**Background:** An erector spinae plane block (ESPB) has gained popularity due to its effectiveness and simplicity for pain relief. However, it is uncertain whether an ESPB provides superior analgesia after a VATS or thoracotomy compared to other regional and systemic analgesic techniques. **Methods:** A retrospective cohort study was conducted from January to June 2023 comparing an ESPB with intravenous combination analgesia (IV–CA) in VATS patients and with thoracic epidural analgesia (TEA) in thoracotomy patients. The primary endpoint was the opioid demand during the first two hours in the post-anesthesia care unit (PACU). The secondary outcomes included the pain scores and adverse events. **Results:** A total of 61.2% of the 165 included VATS patients and 56.9% of the 72 thoracotomy patients were treated with an ESPB. Following a VATS, an ESPB decreased the median piritramide demand (7.5 [3.0 to 12.0] vs. 10.5 [6.5 to 15.5] mg, *p* < 0.01). However, after a thoracotomy, an ESPB increased the median piritramide demand (12.0 [6.0 to 15.0] vs. 3.0 [0.0 to 9.0] mg, *p* < 0.01). The pain scores and adverse events were similar between the groups. **Conclusions:** An ESPB reduces the piritramide demand in VATS patients compared with IV–CA, providing similar pain relief. However, in thoracotomy patients, an ESPB is associated with an increased piritramide demand compared to TEA. An ESPB is an attractive add-on to IV–CA after a VATS, while TEA remains the gold standard after a thoracotomy.

## 1. Introduction

An ESPB [1] is a regional anesthesia technique with growing popularity due to its simplicity. An ESPB is increasingly being used in patients undergoing a VATS, mini-thoracotomy, general surgery, or orthopedic surgery [2,3,4,5,6,7]. It involves the ultrasound-guided injection of local anesthetic into the fascial plane beneath the erector spinae muscle [1,8]. The method time-sparingly provides effective analgesia for surgeries involving the chest wall [8,9,10,11,12,13].

Regional anesthesia is a key component of early recovery after surgery (ERAS) programs in thoracic surgery, reducing postoperative complications and the hospital length of stay (LOS) by providing opioid-sparing analgesia in and after the post-anesthesia care unit (PACU) [14,15,16,17]. An ESPB may significantly reduce postoperative opioid requirements [9,10,11,18,19], potentially decreasing opioid-related side effects such as nausea, vomiting, and respiratory depression. However, these small studies may have been underpowered to detect differences in all the outcomes. A recent review by La Via et al. showed that that an ESPB is safe, is simple, and provides sufficient somatic and visceral analgesia after a robotic-assisted thoracic surgery (RATS) [13]. Durey et al. found that an ESPB was associated with less postoperative static and dynamic pain when compared to a paravertebral block (PVB) [20]. Finnerty et al. found improved analgesia and recovery after an ESPB compared to a serratus anterior plane block (SAPB) after an RATS [21]. Investigations into the effectiveness of an ESPB in patients undergoing a VATS [10,11,18,19,22,23] or a thoracotomy [2,9,24,25] used various perioperative analgesic regimens, yielding conflicting results, and only one study compared an ESPB to a combination of multiple non-opioids and opioids for analgesia (IV-CA) [23].

We investigated the effectiveness of an ESPB in a conveniently sized cohort of patients, comparing it to IV-CA in VATS patients and TEA in thoracotomy patients. We hypothesized that an ESPB would reduce the early postoperative opioid demand compared to IV-CA after a VATS and that an ESPB would be non-inferior to TEA after a thoracotomy.

## 2. Materials and Methods

### 2.1. Design

A retrospective single-center study was performed in patients undergoing a VATS or thoracotomy from January to July 2023 in a tertiary hospital. The study protocol was approved by the institutional review board (EC number: 2204/2023).

### 2.2. Patients

We included consecutive patients (i) aged 18 years or older (ii) who underwent a VATS or thoracotomy (iii) admitted to the PACU after surgery. We excluded patients who required postoperative prolonged intubation or underwent other regional anesthesia or additional surgical procedures. We also excluded patients on opioids prior to surgery, those who received opioids other than piritramide postoperatively, those who were allergic to piritramide, or those with missing data regarding pain management.

### 2.3. ESPB

At our center, an ESPB is performed under general anesthesia in the lateral decubitus position immediately after surgery. After locating the transverse process, T4–6, depending on the level of the thoracotomy, and achieving good visual conditions of the superficial muscles using a 13–6 MHz linear array US transducer (HFL38xi, Sonosite, Bothell, WA, USA) in a cephalad-to-caudal direction, a 70 mm 22-gauge needle (USC 70 Evolution, Felsberg, Germany) is inserted using the in-plane approach. After confirming an optimal needle tip position by hydrodissection, an interfascial injection of 20 to 30 mL of 0.375% ropivacaine (Sintetica, Münster, Germany) combined with 0.5–1 µg.kg^–1^ patient body weight of dexmedetomidine (Orion Pharma Austria, Vienna, Austria) is performed. Systematic training of the ESPB technique is conducted at our department and all the ESPBs were performed by or under the direct supervision of four experienced consultants to ensure a high level of standardization.

### 2.4. Standard Analgesia

At our center, during general anesthesia, remifentanil and fentanyl are combined with propofol and rocuronium. In patients managed with remifentanil, a bolus of fentanyl (1 to 2 µg.kg^−1^ body weight) is administered at the end of the rocuronium infusion, with no other long-acting opioids used intraoperatively. In patients with extended resections via a thoracotomy, a thoracic epidural catheter (Portex Epidural Catheter, Smiths Medical, Minneapolis, MN, USA) is inserted 3 to 5 cm into the epidural space between Th 6 and Th 9 before the surgery. After excluding the intrathecal position by injecting a test dose (2 to 3 mL of Xylocaine 2%, Recipharm, Monts, France), 5 to 8 mL of 0.2 mg.mL^−1^ ropivacaine is injected and an adequate block is confirmed by the loss of temperature discrimination to cold. For IV-CA, two of the following non-opioids (metamizole (G.L. pharma, Lannach, Austria), acetaminophen (Industria Farmaceutica Galenica, Monteroni d’Arbia, Italy), diclofenac (Fresenius Kabi, Seiersberg, Austria)) are supplemented with piritramide (HBM Pharma, Martin, Slovakia) in weight-adjusted doses (1.5 to 4.5 mg i.v.) if the visual analogue scale (VAS) score exceeds 3 in the PACU.

### 2.5. Data Collected

We collected demographic data and baseline characteristics for each patient, including their gender, age, American Society of Anesthesiology (ASA) score, and primary diagnosis. We recorded the postoperative piritramide demand, analgesia failure as defined by a VAS score ≥ 7, the type and dosage of intraoperative opioids, and the intra- and postoperative nonopioid analgesics. Pain levels were assessed routinely every 30 min until discharge from PACU by nursing staff using the VAS score from 0 to 10, and the sedation levels were assessed using the modified Ramsay sedation scale (mRSS). Patients with missing primary outcome data were excluded from the analysis after screening.

### 2.6. Definitions

Analgesia failure was defined as a median VAS score ≥ 7 during the first postoperative hour. The total anesthesia time encompassed the cumulative duration of anesthesia induction and emergence from anesthesia after skin closure. All the data were extracted from our automated patient data management system.

### 2.7. Study Endpoints

The primary endpoint was the cumulative postoperative piritramide demand during the first 2 h after arrival in the PACU. Piritramide is the standard intravenous medication for pain treatment at our department. The therapeutic aim for postoperative pain therapy after a thoracic surgery is to achieve a VAS score < 3. Piritramide is titrated until this therapeutic aim is reached. Therefore, we chose the amount of piritramide as the primary outcome measure. The secondary endpoints included the pain scores and the dosage of intraoperative opioids, postoperative opioids, and non-opioids. We also reported the length of stay (LOS) in the operating room and the PACU, the total anesthesia time, and adverse events.

### 2.8. Sample Size Calculation

We did not perform a formal sample size calculation. With a turnover of 450 patients undergoing a VATS per year, we assumed that a 6-month-long period would be sufficient to achieve a meaningful comparison between patients who underwent an ESPB and those who did not.

### 2.9. Statistical Analysis

We divided the cohort into two groups: patients undergoing a VATS and those undergoing thoracotomies. We compared an ESPB to IV-CA after a VATS and an ESPB to TEA after a thoracotomy. The variables are presented using counts and percentages, means and standard deviations (SDs), or medians and interquartile ranges (IQRs), depending on the data distribution.

To evaluate the primary and secondary outcomes, we employed a two-sided superiority hypothesis. The mean differences in the postoperative piritramide demand between patients with an ESPB and those receiving IV-CA after a VATS, and between patients with an ESPB and those receiving TEA after a thoracotomy, are reported.

We performed a post hoc analysis that included patients with a prolonged stay in the comparing cumulative piritramide demand in the first 6 h. We also descriptively compared the costs for single-use equipment and medication for an ESPB and TEA. A post hoc sample size calculation was performed to assess the sample’s adequacy for detecting the primary outcome.

All the statistical analyses were performed using SPSS for Windows, version 29.0 (IBM Corp., Armonk, NY, USA), and GraphPad Prism for Windows, version 10.0.0 (GraphPad Software, Boston, MA, USA). We considered a *p*-value of less than 0.05 as statistically significant.

## 3. Results

### 3.1. Patients

From January to July 2023, 207 patients underwent a VATS and 86 underwent thoracotomies. The main reasons for exclusion from the analysis were an intraoperative intercostal block performed during a VATS and remaining intubated after a thoracotomy surgery (Figure 1). Of the 165 included VATS patients, 101 (61.2%) underwent an ESPB. Among the 72 included thoracotomy patients, 41 (56.9%) underwent an ESPB. The majority of the patients were male, had a malignancy, and underwent elective surgery (Table 1). In the VATS group, the procedures (segment resection, lobectomy, etc.) did not differ between patients with and without an ESPB. In the thoracotomy group, extensive tumor resection occurred mainly in the TEA group (Table 1).

### 3.2. Postoperative Piritramide

After a VATS, an ESPB reduced the median postoperative piritramide demand compared to IV-CA (7.5 [3.0 to 12.0] vs. 10.5 [6.5 to 15.5] mg; mean difference: –3.4 (1.0) mg; *p* < 0.01). In contrast, after a thoracotomy, an ESPB increased the median postoperative piritramide demand compared to TEA (12.0 [6.0 to 15.0] vs. 3.0 [0.0 to 9.0] mg; mean difference: +6.9 (1.5) mg; *p* < 0.01; Figure 2A,C and Table 2).

### 3.3. Postoperative Pain Scores

The VAS score during the first 2 h in the PACU was comparable in all groups, irrespective of the surgical approach or the method of analgesic treatment (Figure 2B,D and Table 2).

### 3.4. Exploratory Endpoints

In VATS patients, a prolonged LOS in the OR and prolonged anesthesia time were observed in patients with an ESPB compared to IV-CA (Table 3). In thoracotomy patients, the LOS in the PACU and the hospital stay were shorter in patients with an ESPB compared to patients with TEA (Table 3).

### 3.5. Adverse Events

There were no complications associated with an ESPB or TEA. There were no differences in adverse events after a VATS, while after a thoracotomy, patients with an ESPB required antiemetic drugs more often (Table 4).

### 3.6. Post Hoc Analyses

The findings regarding opioid consumption were consistent with the main analysis (Appendix A). In VATS patients, the pain scores were lower in patients with an ESPB compared to patients with IV-CA (Figure 2C,D). The post hoc sample size calculation indicated that the cohort size was adequate for detecting a significant difference in early postoperative piritramide, with a power of 0.91 in the VATS group and 0.99 in the thoracotomy group.

Our cost analysis showed moderate costs of EUR 69.60 per ESPB and EUR 47.20 per TEA, including the material and medication for a 24 h period, but not including the costs for personnel, reusable medical equipment, opioids, and non-opioids (Appendix A).

## 4. Conclusions

Our findings can be summarized as follows: (1) in VATS patients, an ESPB was associated with a reduction in the postoperative opioid demand during the initial two hours in the PACU compared to IV-CA; (2) in contrast, in thoracotomy patients, an ESPB was associated with an increase in the early postoperative opioid demand compared to patients with TEA; (3) in both groups, pain management with an ESPB vs. IV-CA or an ESPB vs. TEA was equally effective at achieving acceptable VAS values; (4) there were no major differences in adverse events for IV-CA, an ESPB, or TEA; and (5) the prolongation of anesthesia time through an ESPB was minimal. 

Our data strongly support an ESPB as a safe, fast, and easy-to-perform analgetic approach in patients undergoing a VATS [26]. The reduction in the opioid demand may lead to reduced opioid-associated adverse effects in the postoperative period and provide increased patient safety. Furthermore, increasing numbers of new minimally invasive operative techniques demand the implementation of modern ultrasound-guided analgetic techniques like ESPBs to provide lasting, effective, and safe pain relief.

We included a relatively large number of patients undergoing a VATS or a thoracotomy. Consecutive patients were included during a predefined time period, with the exclusion criteria limited to those receiving an additional anesthesia technique or remaining intubated. The prespecified analysis plan was strictly adhered to after cleaning the database. A standardized evaluation of the pain and sedation scores was performed and documented by the nursing team. The nurses were unaware of the study, ruling out documentation and treatment bias.

Several investigations have examined the use of an ESPB for pain management after a VATS and reported beneficial effects. The majority did not specify or describe the use of IV-CA, which combines multiple non-opioids with opioids [10,11,18,27]. Our findings support these results, showing a moderate reduction in the piritramide demand after a VATS compared to IV-CA. However, this effect has not been consistently reported [22]. A potential reason is that the opioid-sparing effects reported by previous authors [10,11,18,27] were too moderate to be detected [22]. An ESPB was not associated with a reduction in the PACU or hospital LOS. These endpoints may be influenced by factors beyond analgesic management [14,18].

We observed no benefit of an ESPB after a thoracotomy compared to TEA. This confirms TEA as the gold standard for post-thoracotomy pain management [28]. A previous study reported an equal effectiveness for an ESPB compared to TEA [25,29]. This discrepancy may be due to the exclusion of patients with ESPB failures, while not excluding patients with TEA failure [29]. In our study, patients were preselected for TEA when extensive and painful surgery was expected. This preselection, which was unfavorable for the TEA group, may explain why there was no difference in the pain scores between the TEA and ESPB groups, but TEA was superior with respect to the postoperative opioid demand. The same applies for the age difference between the patients in the thoracotomy group. The patients in the TEA group were younger than the patients in the ESPB group. Since pain is perceived as more severe in younger patients compared to older patients [30,31], this fact is unfavorable for the TEA group. However, the fact that the postoperative opioid demand was significantly decreased in the TEA group, despite the patients being younger and undergoing more extensive surgery, only underscores the effectiveness of TEA.

Another factor that may explain the superiority of TEA over an ESPB after a thoracotomy is the variable spread of the local anesthetic agent in an ESPB. Local anesthetic spread may differ significantly between patients [4,5,6], leading to an inconsistent area of anesthesia. This may be an explanation for the contradicting results in previous works. While the injected volume has been shown to impact the spread of local anesthetic, an ESPB is effective in both the sitting position and the lateral decubitus position [10,18,26,32]. Furthermore, an ESPB was compared to TEA in our study, which is the golden standard of postoperative pain therapy after a thoracotomy, despite its challenges [13]. The results may have been different if an ESPB had been compared to intravenous analgesia only. However, all of our thoracic patients receive an ESPB or TEA if extubation is planned.

The ultimate goal for postoperative pain management is to control pain and discomfort to support speedy recovery. The perceived levels of pain were similar and the aim of effective pain control was achieved in all patients. Previous papers demonstrating a significant reduction in the VAS score with an ESPB differ from our findings, as they compared an ESPB with various analgesia regimens [9,10,11,18]. In our study, all the patients received the same IV-CA regimen.

During the implementation of an ESPB in our department, we faced two significant challenges. One of them was the profound understanding of thoracic anatomy, particularly in the lateral decubitus position. Therefore, four experienced anesthesia consultants performed or directly supervised the ESPBs. The other significant challenge we faced was the prolongation of anesthesia time. If an ESPB is performed preoperatively, a significant disruption of the operative work flow in the OR may occur [13]. By performing the procedure immediately after skin closure and extending the sterile operative field down to the vertebral column, the time for the ESPB was minimized. Ultrasound with high-resolution linear probes must be readily available at the time of skin closure to avoid the unnecessarily disruption of work flow in the OR.

No adverse effects of ESPBs were observed in our study, which is encouraging and consistent with the literature, indicating a lack of severe complications [9,10,11,18,32,33]. Isolated case reports of pneumothorax, hematomas, and local anesthetic systemic toxicity have been documented [34,35,36], but were not observed in our study.

Our findings suggest that, in patients undergoing a VATS, an ESPB is a safe way to reduce the postoperative opioid demand compared to using IV-CA. Based on these results, we have integrated ESPBs into our standard pain management protocol for VATS procedures, complementing the existing i.v. combination analgesia protocol to enhance patient recovery and comfort. For patients undergoing extensive resections via a thoracotomy, we continue to use TEA as our standard of care, while—given the results of multiple reviews—recognizing the possible role of an ultrasound-guided paravertebral block for these patients, which may avoid some of the undesired effects of TEA [13,24,26].

Our study has some limitations, including the retrospective design and the relatively short 2-hour observation period for the primary outcome. As a result of the retrospective design, the patient numbers and types of surgical procedures between groups differed, both after an ESPB and after a VATS. Since patients were preselected to receive an ESPB, IV-CA, or TEA, based on patient factors and operative procedures, there was the risk of over- or underestimating the effect size of an ESPB.

The aforementioned relatively short observation period was chosen for three reasons. Many patients were transferred to the ward after two hours if they were well. Second, pain monitoring and piritramide administration are highly standardized at our PACU, more so than after transfer to the surgical ward, providing a high validity. Third, during the initial two hours in the PACU, the effect of an ESPB should be especially pronounced, thus best reflecting the effectiveness during the early postoperative period. While 2 h is a short period, pain reduction early after surgery is imperative to reduce myocardial stress and chronic postoperative pain caused by inadequate analgesia, improve the effective cough, and reduce opioid-associated side effects [16,17]. Notably, the results of our post hoc analysis, which included observations up to 6 h, were consistent with the findings of the primary analysis. The PACU nurses were not blinded, but since this was a retrospective analysis, they were unaware of the study.

Another limitation of this study is the lack of detailed monitoring of the area of analgesia after the ESPB. Our department’s standard of care is that an ESPB is performed at the end of the surgical procedure, while patients are still under general anesthesia. A detailed analysis of the spread of analgesia, determined by the loss of sensation to hot or cold, is usually not performed in the PACU.

Our post hoc cost analysis included the costs per block or TEA; it did not, however, include the costs of intravenous opioids, non-opioids, or personnel that arise during the preoperative performance of TEA or postoperative patient visits.

Based on these results, we concluded that an ESPB decreases the early postoperative cumulative piritramide demand in VATS patients and, thus, is likely to reduce opioid side effects. However, TEA was superior to an ESPB at reducing the piritramide demand in thoracotomy patients, thus underpinning the role of TEA for thoracotomies.

## Figures and Tables

**Figure 1 jcm-14-02870-f001:**
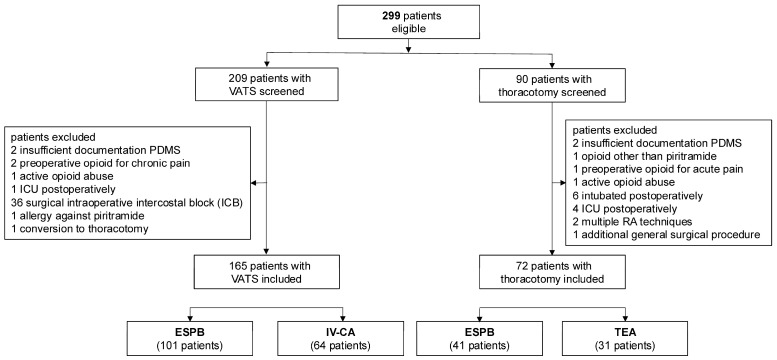
Flowchart of patients. Abbreviations: VATS, video-assisted thoracic surgery; ESPB, erector spinae plane block.

**Figure 2 jcm-14-02870-f002:**
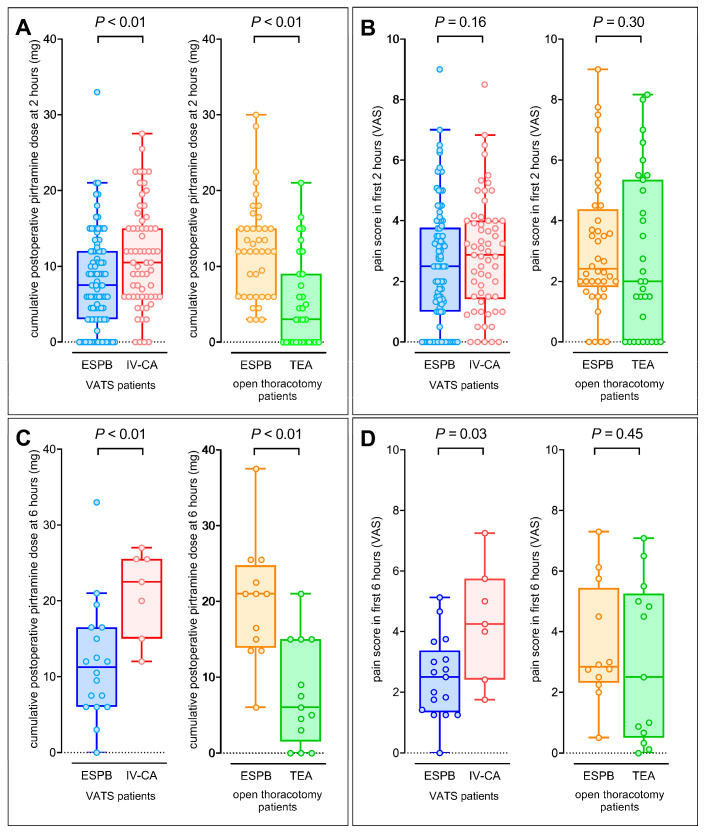
Box and whisker plots of postoperative piritramide dose with and without ESPB and pain scores. (**A**) Piritramide dose and pain scores in patients after ESPB or (**B**) thoracotomy within 2 h after surgery; (**C**,**D**) within 6 h after surgery. Abbreviations: VATS, video-assisted thoracic surgery; ESPB, erector spinae plane block; IV-CA, intravenous combined analgesia; TEA, thoracic epidural analgesia.

**Table 1 jcm-14-02870-t001:** Patient demographics and baseline characteristics.

	VATS Patients (N = 165)		Thoracotomy Patients (N = 72)			
	With ESPB (N = 101)	With IV–CA (N = 64)	*p*	With ESPB (N = 41)	With TEA (N = 31)	*p*
age, years	60.9 (17.5)	60.4 (50.7)	0.84	63.9 (10.9)	52.7 (19.6)	<0.01
sex, female	46 (45.5)	27 (42.2)	0.67	15 (36.6)	10 (32.3)	0.70
BMI, mean (SD)	25.7 (6.1)	26.2 (6.4)	0.65	27.3 (5.4)	23.8 (4.2)	<0.01
ASA score	3.0 [2.0 to 3.0]	3.0 [2.0 to 3.0]	0.90	3.0 [2.0 to 3.0]	3.0 [2.0 to 3.0]	0.42
type of surgery			0.12			0.38
elective surgery, *n* (%)	91 (90.1)	57 (89.1)		40 (97.6)	30 (96.8)	
urgent surgery, *n* (%)	10 (9.9)	7 (10.9)		1 (2.4)	1 (3.2)	
analgesia						
ESPB, *n* (%)	101 (100.0)	N.A.		41 (100.0)	N.A.	
before surgery	5 (5.0)	N.A.		0 (0.0)	N.A.	
TEA	0 (0.0)	0 (0.0)		0 (0.0)	31 (100.0)	
before surgery	N.A.	N.A.		N.A.	31 (100.0)	
primary diagnosis			0.84			0.04
malignancy	68 (67.3)	29 (45.3)		35 (85.4)	18 (58.1)	
pathology pending	33	19		8	3	
adenocarcinoma	17	2		7	4	
squamous cell carcinoma	5	4		7	5	
non-small cell lung cancer	3	1		1	1	
metastasis	8	2		7	2	
other malignancy	2	1		5	3	
pneumothorax	12 (11.9)	9 (14.1)		0 (0.0)	0 (0.0)	
pleural effusion	8 (7.9)	10 (15.6)		0 (0.0)	0 (0.0)	
hemothorax	0 (0.0)	5 (7.8)		0 (0.0)	0 (0.0)	
empyema	5 (5.0)	4 (6.3)		6 (14.6)	2 (6.5)	
emphysema	1 (1.0)	1 (1.6)		0 (0.0)	0 (0.0)	
pectus deformity	0 (0.0)	0 (0.0)		0 (0.0)	5 (16.1)	
other	7 (6.9)	6 (9.4)		0 (0.0)	6 (19.3)	
surgical intervention						
EASR/ASR, *n* (%)	34 (33.7)	22 (34.4)		10 (24.4)	1 (3.2)	
single EASR, *n*	15	12		6	1	
EASR + ASR, *n*	0	1		0	0	
multiple EASR, *n*	8	6		0	0	
multiple EASR + ASR, *n*	1	0		0	0	
single ASR, *n*	7	3		4	0	
multiple ASR, *n*	3	0		0	0	
lobectomy/pneumonectomy, *n* (%)	44 (43.6)	14 (21.9)		26 (63.4)	13 (41.9)	
lobectomy, *n*	21	6		11	4	
lobectomy + EASR, *n*	6	1		2	2	
bilobectomy, *n*	1	0		1	0	
pleurectomy, *n*	10	5		5	3	
pleurectomy + EASR, *n*	6	2		0	0	
sleeve bi-lobectomy	0	0		1	1	
sleeve lobectomy	0	0		3	1	
pneumonectomy	0	0		3	2	
pleurodesis	5 (5.0)	5 (7.8)		0 (0.0)	0 (0.0)	
pleurodesis	4	5		0	0	
pleurodesis + IPC	1	0		0	0	
VATS, *n* (%)	11 (10.9)	5 (7.8)		0 (0.0)	0 (0.0)	
diagnostic VATS	6	3		0	0	
diagnostic VATS + IPC	3	2		0	0	
diagnostic VATS + pleurectomy	1	0		0	0	
diagnostic VATS + multiple EASR	1	0		0	0	
miscellaneous, *n* (%)	7 (6.9)	11 (17.2)		6 (14.6)	17 (54.8)	
hematoma evacuation	0	2		1	0	
minor tumor resection	0	0		0	2	
extensive tumor resection	0	0		1	7	
Nuss operation	0	0		0	4	
other	7	9		3	4	

Data are presented as medians with interquartile ranges [IQRs]; VAS used a scale of 1 to 10. Abbreviations: VAS, visual analogue scale; PACU, post-anesthesia care unit; VATS, video-assisted thoracic surgery; ESPB, erector spinae plane block; IV–CA, intravenous combination analgesia; TEA, thoracic epidural analgesia.

**Table 2 jcm-14-02870-t002:** Piritramide dose and VAS score for pain in first 2 h after arrival to PACU.

	VATS Patients (N = 165)			Thoracotomy Patients (N = 72)		
	With ESPB (N = 101)	With IV–CA (N = 64)	*p*	With ESPB (N = 41)	With TEA (N = 31)	*p*
Primary endpoint						
Median piritramide dose in first 2 h, mg [IQR]	7.5 [3 to 12]	10.5 [6.5 to 15.5]	<0.01	12.0 [6.0 to 15.0]	3.0 [0.0 to 9.0]	<0.01
Secondary endpoint						
Median VAS for pain in first 2 h, score [IQR]	2.5 [1.0 to 3.8]	2.9 [1.5 to 4.0]	0.19	2.4 [1.8 to 4.4]	2.0 [0.0 to 5.4]	0.30

Data are presented as medians with interquartile ranges [IQRs]; VAS used a scale of 1 to 10. Abbreviations: VAS, visual analogue scale; PACU, post-anesthesia care unit; VATS, video-assisted thoracic surgery; ESPB, erector spinae plane block; IV–CA, intravenous combination analgesia; TEA, thoracic epidural anesthesia.

**Table 3 jcm-14-02870-t003:** Sedation score, LOS, anesthesia time, and intraoperative and postoperative medication.

	VATS Patients (N = 165)			Thoracotomy Patients (N = 72)		
	With ESPB (N = 101)	With IV–CA (N = 64)	*p*	With ESPB (N = 41)	With TEA (N = 31)	*p*
sedation score, median [IQR]						
first hour	3.0 [2.5 to 3.0]	3.0 [2.4 to 3.0]	0.74	3.0 [2.5 to 3.0]	3.0 [3.0 to 3.0]	0.18
second hour	3.0 [2.5 to 3.0]	3.0 [2.5 to 3.0]	0.91	2.7 [2.5 to 3.0]	2.8 [2.5 to 3.0]	0.57
third hour	3.0 [2.5 to 3.0]	3.0 [2.5 to 3.0]	0.81	3.0 [2.5 to 3.0]	3.0 [3.0 to 3.0]	0.18
LOS, median [IQR]						
in OR, min	178.0 [132.0 to 229.0]	121.0 [98.6 to 195.5]	<0.01	225.0 [136.0 to 290.0]	232.0 [206.0 to 309.0]	0.85
anesthesia time, min	33.0 [25.0 to 40.5]	30.0 [22.3 to 37.5]	<0.01	34.0 [31.0 to 41.0]	35.0 [25.0 to 54.0]	0.64
in PACU, min	135.0 [110.0 to 180.0]	125.0 [106.5 to 180.0]	0.71	150.0 [117.0 to 832.0]	210.0 [165.0 to 1095.0]	0.02
in hospital, days	4.0 [3.0 to 5.0]	3.0 [2.8 to 6.0]	0.61	5.0 [3.5 to 7.5]	6.0 [5.0 to 11.0]	0.03
analgesia						
intraoperative:						
remifentanil, *n* (%)	95 (94.1)	58 (89.1)	0.25	39 (95.1)	27 (87.1)	0.43
fentanyl, mg, median [IQR]	300 [200 to 388]	300 [200 to 400]	0.09	300 [250 to 425]	300 [200 to 400]	0.54
intra- and postoperative:						
metamizole, *n* (%)	85 (84.2)	55 (85.9)	0.76	35 (85.4)	28 (90.3)	0.53
metamizole, mg, median [IQR]	2000 [1000 to 2500]	1000 [1000 to 2500]	0.46	2000 [1000 to 2500]	1000 [1000 to 2000]	0.11
diclofenac, *n* (%)	55 (54.5)	31 (48.4)	0.45	28 (68.3)	15 (48.4)	0.09
diclofenac (mg)	0.0 [0.0 to 75.0]	75.0 [0.0 to 75.0]	0.32	75.0 [0.0 to 75.0]	0.0 [0.0 to 75.0]	0.11
acetaminophen, *n* (%)	38 (37.6)	29 (45.3)	0.33	16 (39.0)	6 (19.4)	0.07
acetaminophen, mg, median [IQR]	0.0 [0.0 to 1000.0]	0.0 [0.0 to 1000.0]	0.46	0.0 [0.0 to 1000.0]	0.0 [0.0 to 0.0]	0.07

The sedation score was measured on a scale between 0 and 3. Abbreviations: LOS, length of stay; OR, operating room; PACU, post-anesthesia care unit; VATS, video-assisted thoracic surgery; ESPB, erector spinae plane block; IV–CA, intravenous combination analgesia; TEA, thoracic epidural anesthesia.

**Table 4 jcm-14-02870-t004:** Adverse events.

	VATS Patients (N = 165)			Thoracotomy Patients (N = 72)		
	With ESPB (N = 101)	With IV–CA (N = 64)	*p*	With ESPB (N = 41)	With TEA (N = 31)	*p*
adverse events						
any event, *n* (%)	29 (28.7)	20 (31.7)	0.68	41 (100.0)	22 (71.0)	<0.01
hospital mortality, *n* (%)	0 (0.0)	2 (3.1)	0.07	0 (0.0)	1 (3.2)	0.25
antiemetic medication, *n* (%)	26 (25.7)	15 (23.7)	0.74	41 (100.0)	22 (71.0)	<0.01
motor block, *n* (%)	0 (0.0)	0 (0.0)	–	0 (0.0)	0 (0.0)	–
analgesia failure, *n* (%)	8 (7.9)	5 (7.8)	0.98	6 (14.6)	5 (16.1)	0.86
local anesthetic toxicity, *n* (%)	0 (0.0)	0 (0.0)	–	0 (0.0)	0 (0.0)	–
delirium, *n* (%)	1 (1.0)	0 (0.0)	1.0	1 (2.4)	0 (0.0)	0.63

Abbreviations: PACU, post-anesthesia care unit; VATS, video-assisted thoracic surgery; ESPB, erector spinae plane block; IV–CA, intravenous combination analgesia; TEA, thoracic epidural anesthesia; BGA, blood gas analysis; PaO_2_, arterial partial pressure of oxygen; PaCO_2_, arterial partial pressure of carbon dioxide; O_2_ (L/min), supplemental oxygen via Venturi mask.

## Data Availability

The dataset of the current study is available via the corresponding author upon request.

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
