# Peer review of "Effectiveness and Safety of Erector Spinae Plane Block vs. Conventional Pain Treatment Strategies in Thoracic Surgery"

_jcm, 2025, doi:10.3390/jcm14092870_

Round 1

Reviewer 1 Report

Comments and Suggestions for Authors

This is a well-conducted retrospective cohort study examining the effectiveness of ESPB compared to conventional analgesic approaches in thoracic surgery. The manuscript provides valuable insights into postoperative pain management strategies, though there are some areas that could be strengthened.

Major Strengths:

  1. Clear clinical relevance with practical implications for thoracic surgery pain management
  2. Adequate sample size with post-hoc power analysis confirming statistical validity
  3. Well-defined outcome measures and standardized assessment protocols
  4. Balanced presentation of both positive and negative findings
  5. Thorough statistical analysis and appropriate data presentation

Major Concerns:

  1. Short primary outcome window (2 hours) - though addressed in limitations and supported by 6-hour post-hoc analysis
  2. Potential selection bias in thoracotomy group where TEA was preferentially used for more extensive procedures
  3. Lack of standardization in ESPB timing (pre vs post-operative)

Specific Comments:

Introduction:

  • "ESPB [1] is a regional anesthesia technique with growing popularity due to its simplicity. ESPB is increasingly used in patients undergoing various types of surgeries [2–7]." Authors should also mention robotic surgery in this contest. Please briefly discuss doi: 10.3390/jcm13113141

Methods:

  • Consider providing more detail about the ESPB technique standardization across practitioners
  • Clarify the rationale for choosing piritramide as the primary outcome measure
  • Include information about missing data handling

Results:

  • Consider adding subgroup analyses based on surgical complexity
  • Provide more detail about failed blocks or technical difficulties
  • Include cost comparison data if available

Discussion:

  • Expand on clinical implications of findings
  • Address potential mechanisms for differential effectiveness between VATS and thoracotomy
  • Consider discussing implementation challenges

Reviewer 2 Report

Comments and Suggestions for Authors

Review Report: jcm-3535796

Effectiveness and Safety of Erector Spinae Plane Block versus Conventional Pain Treatment Strategies in Thoracic Surgery

This study represents a commendable effort by the authors to investigate the safety and efficacy of ESPB in VATS and thoracotomy surgery compared to iv opioids and TEA, respectively.

Although this is a retrospective study, it is well designed.

It would have been useful if the authors had commented on this:

  1. The difference in the VATS group between the number of participants in the ESP and IV CA groups (101 vs. 64) and how this difference may influence the results. This point could also be mentioned in the limitations of the study.
  2. The statistically important difference regarding the age of the patients in the TEA group.
  3. The different types of surgery performed in each group. For example, there were many more (double) lobectomies performed in the ESP group than in the TEA group, while there were more excessive tumor resections in the TEA group. This is another point that may be worth mentioning in the study's limitations.

After the results, a Discussion of the findings should follow and then conclusions.

The discussion could be enriched with the results of the available reviews and meta-analyses on the use of ESPB in thoracic surgery.

Thank you for your time and effort.
